# Analysis of Steady Groundwater Flow in Confined Aquifer Due to Long-Strip Pit Dewatering with Suspended Cut-Off Wall

**Weijia Tan [1], Haibo Kang [1], Jin Xu [2] and Xudong Wang [1,\*]**

1　Institute of Geotechnical Engineering, Nanjing Tech University, Nanjing 211816, China;
651081401003@njtech.edu.cn (W.T.); 202162126007@njtech.edu.cn (H.K.)
2　School of Civil Engineering, Yantai University, Yantai 264005, China; jinxu@ytu.edu.cn
\*　Correspondence: cewxd@njtech.edu.cn

**Abstract:** This study investigates the steady flow resulting from dewatering by a partially penetrating well in a confined aquifer with a cut-off barrier. By considering flow in both horizontal and vertical directions and incorporating the barrier and pumping well as flow boundary conditions, separate mathematical models are established for the inside and outside of the cut-off barrier. The interaction between these zones is ensured through continuous conditions along the opening of the two zones. A semi-analytical solution is derived for the problem using the finite Fourier cosine transform and boundary transformation methods. The effectiveness of the method is verified by comparing it with the finite element numerical results and pumping test data respectively. Based on the proposed solutions, we proceed to analyze the influence of some relevant factors: the extent to which the cut-off wall is embedded within the confined aquifer, the depth of the partially penetrating well, and the distance to the lateral head boundary. Results indicate that a greater depth of the cut-off wall leads to a reduced pumping rate requirement for achieving a desired drawdown of the confined water level within the excavation. According to the presented solution, placing pumping wells near the top of the confined aquifer in excavation dewatering projects can facilitate a faster reduction of the confined water head at the excavation bottom. Additionally, proximity of lateral head boundary could significantly impact dewatering, with closer boundaries reducing dewatering effectiveness due to improved aquifer recharge. Finally, the use of the Fourier method showcases impressive convergence properties in the approach presented in this study. The computed results maintain a high level of approximation quality, even with extremely coarse discretization.

**Keywords:** foundation pit dewatering; cut-off curtain; analytical method; partially penetrating wells

## 1. Introduction

High groundwater levels can pose a significant challenge during the construction of underground structures, particularly when an excavation pit is required [1,2].When the excavation pit is located below the groundwater level, it becomes imperative to actively lower the groundwater level within the underlying aquifer [3–5]. However, as groundwater is pumped from underground aquifers, the consequent reduction in pore pressure can lead to the compaction of underlying sediments. Consequently, insufficiently designed dewatering schemes can trigger the emergence of land subsidence hazard, which holds significant implications for sustainability across various sectors.

Analytical models from groundwater hydraulics have been used for many years in the analysis and design of pit dewatering problems. One of the earliest and most widely used analytical methods is Dupuit's equation [6,7], which assumes that the flow is steady-state, two-dimensional, and confined to a horizontal plane. Another analytical method commonly used in pit dewatering studies is Theis's equation, which is based on the assumption of radial symmetry and homogeneous aquifer properties [8].

From a sustainability standpoint, the utilization of a waterproof wall is a common strategy employed to improve the efficiency of hydraulic level reduction while simultaneously mitigating environmental consequences, such as land subsidence associated with dewatering [9–13]. However, the inclusion of a waterproofing barrier adds further intricacies, introducing factors such as heterogeneity and irregular geometry, which can make conventional analytical approaches inapplicable. Consequently, in order to comprehend the seepage field in proximity to the excavation pit when a cut-off wall is installed during dewatering, a numerical approach, such as the finite element method or finite difference method, is typically required [14–19].

Although numerical simulation methods are widely employed to assess water level variations inside and outside an excavation pit when a cut-off wall is utilized, they may not be the most suitable approach for several reasons. Firstly, the outcomes of numerical analysis heavily rely on the rational selection of parameters and do not directly unveil the underlying mechanisms [20,21]. Secondly, computational modeling in numerical analysis demands a considerable amount of time, posing challenges for engineers seeking to employ it directly for on-site problem-solving [22,23]. In contrast, analytical methods, despite their assumptions and inherent limitations, offer the advantage of directly elucidating the underlying mechanisms, making them suitable for preliminary problem-solving purposes, such as in the phase of engineering designs for the cut-off wall and pumping rates. Pujades et al. developed a semi-empirical equation for estimating the groundwater head difference between the two sides of blocking structures [24]. Shen et al. proposed a simplified analytical method for determining the hydraulic head difference between the two sides of a barrier in the confined aquifer [22]. By a similar strategy, Lyu et al. presented a simple equation to calculate groundwater heads distribution inside and outside of the excavation pit [25]. Yang et al. developed an analytical solution for unsteady flow due to dewatering in foundation pits with a suspended waterproof [23]. However, these approaches rely on assumptions regarding the seepage directions on both sides of the barrier for mathematical simplicity.

In this study, we investigate the steady flow resulting from dewatering by a partially penetrating well in a confined aquifer with a cut-off barrier, in which we extend our analysis to consider the flow in both horizontal and vertical directions. By incorporating both the barrier and the pumping well as flow boundary conditions, we establish two separate mathematical models for the inside and outside of the cut-off barrier. The interaction between the two zones is achieved by the continuous conditions. By employing the finite Fourier cosine transform and boundary transformation methods, we derive a semi-analytical solution for the problem.

## 2. Mathematical Model

Figure 1 presents a schematic diagram of long-strip pit dewatering in a confined aquifer using a partially penetrating well, which can be described as a two-dimensional groundwater seepage problem in the vertical plane. A cut-off wall, extending to a depth of $l_c$ into the confined layer, is installed around the excavation pit to create a barrier that prevents groundwater from flowing into the pit. Due to the symmetry of the computational model, the analysis focuses on the right half of the incomplete well, as shown in Figure 2. In this figure, point a corresponds to the well's intersection with the aquifer's roof, point b designates the intersection between the cut-off wall and the aquifer's roof, point c denotes the cut-off wall's base, point d represents the point where the vertical extension line of the cut-off wall intersects with the aquifer's bottom, and point e stands for the intersection of the vertical extension line of the well with the aquifer's bottom. The pumped confined aquifer has the thickness of $M$, and the pumping rate is $Q_w$.

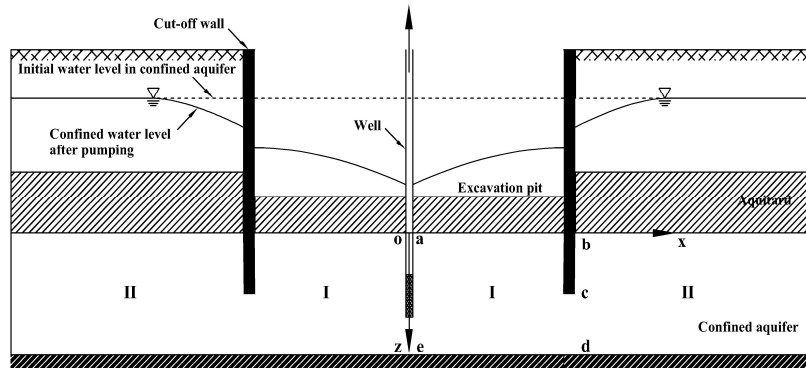

**Figure 1.** Schematic diagram of pit dewatering in confined aquifer with finite lateral dimensions.

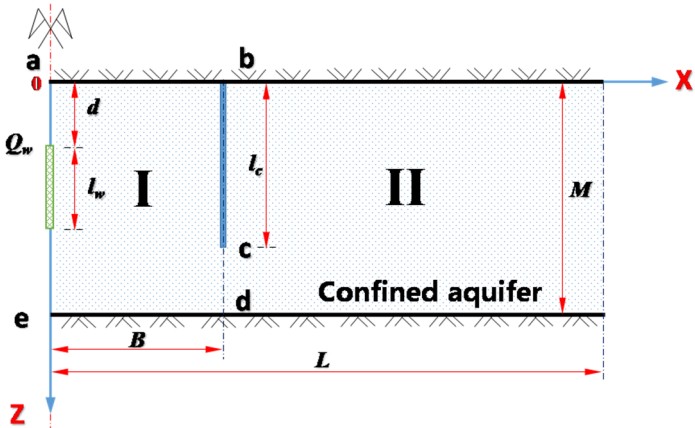

**Figure 2.** Computational model for dewatering in a confined aquifer.

As depicted in Figure 2, the seepage area of the excavation pit can be partitioned into two zones, namely Zone I and Zone II, along the extending axial line of cut-off wall. To establish a coordinate system, we designate the vertical line passing through the center of the excavation pit as the *z*-axis, with the origin defined as the point where the *z*-axis intersects the top of the confined aquifer. This configuration ensures that the well is positioned through the origin of the coordinate system. Furthermore, the following assumptions are incorporated in this study: (a) The outside zone, namely Zone II, has finite lateral dimensions, and its lateral boundary maintains a constant-head condition; (b) the flow in the aquifers follows Darcy's law; (c) the aquifer is homogeneous and the leakage from the upper unconfined layer is negligible; (d) the thickness of the waterproof barrier is being neglected in the study.

### 2.1. Zone I

After considering the lateral boundaries of Zone I, i.e., $x = 0$ and $x = B$, as flux-type conditions, the steady-state groundwater flow in Zone I, considering both the horizontal and vertical directions, can be described by the following boundary value problem [26]:

$$
\begin{cases}
K_x \dfrac{\partial^2 s_I}{\partial x^2} + K_z \dfrac{\partial^2 s_I}{\partial z^2} = 0 \\[2mm]
\left.\dfrac{\partial s_I}{\partial x}\right|_{x=0} =
\begin{cases}
0, & 0 \le z < d \\
-\dfrac{Q_w}{l_w K_x}, & d \le z < d + l_w \\
0, & d + l_w \le z \le M
\end{cases} \\[6mm]
\left.\dfrac{\partial s_I}{\partial x}\right|_{x=B} =
\begin{cases}
0, & 0 \le z < l_c \\
q(z), & l_c \le z \le M
\end{cases} \\[4mm]
\left.\dfrac{\partial s_I}{\partial z}\right|_{z=0} = \left.\dfrac{\partial s_I}{\partial z}\right|_{z=M} = 0
\end{cases}
\tag{1}
$$

where $s_I$ is the unknown hydraulic drawdown in Zone I, which, by definition, is equal to the initial hydraulic level in confined aquifer minus the confined water level after pumping, m; $K_x$ and $K_z$ are horizontal and vertical hydraulic conductivities, m/d; $Q_w$ is the pumping rate per unit width, m$^2$/d; $l_w$ is the length of well screen, m.

In Equation (1), $q(z)$ is an unknown function along the common boundary c-d (in Figure 2) between the zones, which is actually a Neumann boundary condition or flux boundary condition, providing the flux across the c-d segment [27]. Therefore, the unknown $q(z)$ is referred to as the common flux function in this study.

*2.2. Zone II*

In a similar fashion, the steady-state groundwater flow in Zone II can be presented as

$$\left\{ \begin{array}{l} K_x \frac{\partial^2 s_{II}}{\partial x^2} + K_z \frac{\partial^2 s_{II}}{\partial z^2} = 0 \\ \left. \frac{\partial s_{II}}{\partial x} \right|_{x=B} = \left\{ \begin{array}{ll} 0, & 0 \le z < l_c \\ q(z), & l_c \le z \le M \end{array} \right. \\ s_{II}|_{x=L} = 0 \\ \left. \frac{\partial s_{II}}{\partial z} \right|_{z=0} = \left. \frac{\partial s_{II}}{\partial z} \right|_{z=M} = 0, \end{array} \right\} \tag{2}$$

where $s_{II}$ is the hydraulic drawdown in Zone II, which is equal to the initial hydraulic level in confined aquifer minus the confined water level after pumping (see Figure 1).

Additionally, the continuous conditions, including both the hydraulic head and flux, should be satisfied along the common boundary c-d,

$$\left. s_I \right|_{\substack{x = B \\ l_c \le z \le M}} = \left. s_{II} \right|_{\substack{x = B \\ l_c \le z \le M}} \tag{3}$$

$$\left. \frac{\partial s_I}{\partial x} \right|_{x=B} = \left. \frac{\partial s_{II}}{\partial x} \right|_{x=B} \tag{4}$$

It is noted that the flux boundary conditions imposed along the $x = B$ in Equations (1) and (2) automatically lead to a fulfillment of condition (4).

## 3. Fourier Transform Method

Generally, the original problem is challenging to address using analytical methods due to the presence of heterogeneity and geometric irregularity caused by the waterproof barrier. However, the partition along the barrier b-c introduced in the proceeding section simplifies the situation: (a) both the resulting zones are geometrically regular, i.e., rectangles; and (b) the barrier is treated as a no-flow boundary, which is analytically more manageable than the original scenario where it is heterogeneously embedded in the aquifer [28]. Due to heterogeneity, in the original scenario, the Fourier method cannot be directly applied. However, after partitioning, both resulting zones have regular shapes and can be solved separately using finite Fourier transforms. It is also worth noting that, on the other hand, as is typically expected with analytical methods, it requires certain idealized assumptions. For instance, in this paper, the suspended waterproof barrier is treated as a line segment b-c with no width.

By taking the finite Fourier cosine transform of Equation (1) for Zone I in the $z$ variable, which is defined within the range of 0 to $M$, we obtain

$$\left\{ \begin{array}{l} \frac{\partial^2 \bar{s}_I}{\partial x^2} - \beta_n{}^2 \bar{s}_I = 0 \\ \left. \frac{\partial \bar{s}_I}{\partial x} \right|_{x=0} = -\frac{Q_w}{l_w K_x} \int_d^{d+l_w} \cos\left(\frac{\lambda_n z}{M}\right) \mathrm{d}z = -\frac{Q_w}{K_x} \xi_n \\ \left. \frac{\partial \bar{s}_I}{\partial x} \right|_{x=B} = \int_{l_c}^M q(z) \cos\left(\frac{\lambda_n z}{M}\right) \mathrm{d}z \end{array} \right\} \tag{5}$$

where

$$\frac{K_z}{K_x}\left(\frac{\lambda_n}{M}\right)^2 = \beta_n{}^2,$$

$$\xi_n = l_w \int_d^{d+l_w} \cos\left(\frac{\lambda_n z}{M}\right) dz = \begin{cases} \xi_n = 1, n = 0 \\ \xi_n = \frac{\sin\left(\frac{\lambda_n(d+l_w)}{M}\right) - \sin\left(\frac{\lambda_n d}{M}\right)}{\frac{\lambda_n l_w}{M}}, n \geq 1 \end{cases}$$

with $\lambda_n = n\pi$ and $n$ being the parameter of finite cosine transform.

$$\left. \begin{array}{l} \bar{s} = \int_0^M s \cos\left(\frac{n\pi z}{M}\right) dz = \int_0^M s\,dz, n = 0 \\ \bar{s} = \int_0^M s \cos\left(\frac{n\pi z}{M}\right) dz = \int_0^M s \cos\left(\frac{\lambda_n z}{M}\right) dz, n \geq 1 \end{array} \right\}$$

Accidentally, if we set $n = 0$ in (5), we can obtain a mass conservation relationship between unknown flux $q(z)$ at $x = B$ and the pumping rate of the well located at $x = 0$ given by

$$\int_{l_c}^M q(z)dz = -\frac{Q_w}{K_x}$$

After proceeding as above, the Equation (2) for Zone II can be transformed as

$$\left\{ \begin{array}{l} \frac{\partial^2 \bar{s}_{II}}{\partial x^2} - \beta_n^2 \bar{s}_{II} = 0, \\ \left. \frac{\partial \bar{s}_{II}}{\partial x} \right|_{x=B} = \int_{l_c}^M q(z) \cos\left(\frac{\lambda_n z}{M}\right) dz \\ \bar{s}_{II}|_{x=L} = 0; \end{array} \right\} \tag{6}$$

where $\beta_n^2 = \frac{K_z}{K_x}\left(\frac{\lambda_n}{M}\right)^2$.

## 4. Solutions Procedures

After applying the finite cosine transform, the resulting Equations (5) and (6) are ordinary differential equations (ODEs) with corresponding boundary conditions, which can be solved analytically (see Appendix A). Hence, after taking the inversion transform, we obtain the general solutions for the inside (Zone I) and outside (Zone II) of the foundation pit in real space as follows

$$\begin{array}{l} s_I = \frac{1}{M}\left(-\frac{Q_w}{K_x}x + C\right) + \frac{2}{M}\sum_{n=1}^{\infty}\left[\frac{\cosh(\beta_n(B-x))}{\beta_n \sinh(\beta_n B)}\frac{Q_w \xi_n}{K_x}\right. \\ \left. + \frac{\cosh(\beta_n x)}{\beta_n \sinh(\beta_n B)}\int_{l_c}^M q(z)\cos\left(\frac{\lambda_n z}{M}\right)dz\right]\cos\left(\frac{\lambda_n z}{M}\right) \end{array} \tag{7}$$

$$\begin{array}{l} s_{II} = -\frac{1}{M}(L-x)\int_{l_c}^M q(z)dz \\ - \frac{2}{M}\sum_{n=1}^{\infty}\frac{\sinh(\beta_n(L-x))\cos\left(\frac{\lambda_n z}{M}\right)}{\beta_n \cosh(\beta_n(L-B))}\int_{l_c}^M q(z)\cos\left(\frac{\lambda_n z}{M}\right)dz \end{array} \tag{8}$$

The common flux function $q(z)$ defined on the common boundary c-d as well as unknown coefficient $C$ are still unknown in the solutions (7) and (8). They can be determined by applying the continuous condition (3) to these solutions, which states that the hydraulic drawdown of two zones should remain continuous at the common opening. Hence, when substituting $x = B$ in (7) and (8), we obtain the following relationship

$$\begin{array}{l} \frac{Q_w L}{M K_x} - \frac{C}{M} - \frac{2Q_w}{M K_x}\sum_{n=1}^{\infty}\frac{\xi_n \cos\left(\frac{\lambda_n z}{M}\right)}{\beta_n \sinh(\beta_n B)} \\ = \frac{2}{M}\sum_{n=1}^{\infty}\frac{1}{\beta_n}[\tanh(\beta_n(L-B)) + \coth(\beta_n B)]\cos\left(\frac{\lambda_n z}{M}\right)\int_{l_c}^M q(z)\cos\left(\frac{\lambda_n z}{M}\right)dz \end{array} \tag{9}$$

After a one-dimensional discretization along the common boundary c-d, allowing us to approximate the integrals in above relationship using Riemann sums, we eventually obtain the following linear system for the unknowns $q_i$ and $C$(see Appendix B).

$$\begin{bmatrix} a_{11} & \cdots & a_{1N} & 1 \\ \vdots & \ddots & \vdots & \vdots \\ a_{N1} & \cdots & a_{NN} & 1 \\ \Delta z_1 & \cdots & \Delta z_N & 0 \end{bmatrix} \begin{Bmatrix} q_1 \\ \vdots \\ q_N \\ C/M \end{Bmatrix} = \begin{Bmatrix} b_1 \\ \vdots \\ b_N \\ -Q_w/K_x \end{Bmatrix} \tag{10}$$

Hence, when $q_i$ and $C$ are obtained based on the linear system (10), the hydraulic drawdown on both sides of the excavation can be explicitly expressed using Equations (7) and (8).

## 5. Verification

### 5.1. Comparison with Numerical Solutions

To validate the solutions presented in this paper, i.e., Equations (7), (8), and (10), a comparative analysis is conducted with the finite element numerical results. For modeling purposes, a two-dimensional symmetric model is adopted, taking into account the symmetry of the system. The corresponding boundary conditions are as follows: the left side, top, and bottom of the aquifer are designated as impermeable boundaries, while the right side is specified as a constant head boundary. At the same time, sink/source terms are determined based on the pumping rates and well lengths as described in Equation (1). In order to ensure computational accuracy, the mesh is refined at the left boundary of the model, near the pumping well and cut-off wall, and at the top boundary of the confined aquifer with 8700 elements, as shown in Figure 3. The computational parameters are summarized and listed in Table 1.

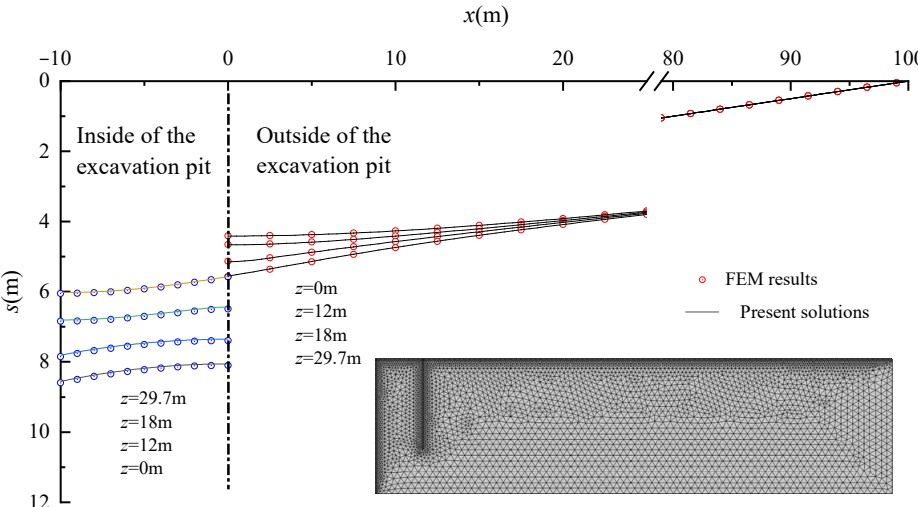

**Figure 3.** Distribution of hydraulic drawdown along the horizontal direction at different depths.

**Table 1.** The calculation parameters in the verification test example.

| Calculation Parameters | Value |
|---|---|
| Depth of cut-off wall insertion $l_c$ | 20 m |
| Thickness of aquifer $M$ | 30 m |
| Spacing between cut-off walls $2B$ | 20 m |
| Pumping rate $Q_w$ | 22.5 m²/d |
| Length of well screen | 15 m |
| Horizontal hydraulic conductivity $K_x$ | 15 m/d |
| Vertical hydraulic conductivity $K_z$ | 15 m/d |

Figure 3 illustrates the distribution of hydraulic drawdown along the horizontal direction at different depths in the confined aquifer zones I and II. A comparison between the analytical and numerical results demonstrates good agreement, validating the accuracy and reliability of the proposed solution for drawdown in an excavation with a suspended cut-off wall. The presented solution in this study aligns well with the dewatering patterns of the excavation site. As the distance from the pumping well to the center of the excavation pit increases, the drawdown becomes more noticeable. Conversely, as the distance from the dewatering well increases, the extent of water level drawdown decreases, gradually approaching zero at the lateral boundary of constant head. The cut-off wall's barrier effect is readily apparent in Figure 3, as illustrated by the discrepancy in drawdown levels. Specifically, within the excavation pit, the drawdown is notably greater compared to the drawdown observed in the aquifer outside the pit. Due to the barrier effect of the cut-off wall, it is also observed that drawdown curves exhibit discontinuity when associated with elevations ($z = 0$ m, $z = 12$ m and $z = 18$ m) above the bottom of the cut-off wall. In contrast, for the elevation below the cut-off wall ($z = 29.7$ m), the drawdown curve remains continuous.

Based on the presented theoretical formula for both Zone I (inside the pit) and Zone II (outside the pit), an equipotential contour of hydraulic drawdown is presented in Figure 4. The figure demonstrates that our results can accurately depict the pattern of hydraulic behavior, providing a precise description of groundwater flow within the vicinity of the cut-off wall. The groundwater flow from the outside aquifer into the pit is impeded by the waterproof barrier. Near the cut-off barrier, the equipotential lines are perpendicular to the cut-off wall, indicating the impermeability of the cut-off wall. Therefore, due to the presence of a cut-off barrier that partially intercepts the aquifer, the seepage path is obstructed, resulting in a blocking effect on groundwater flow: the groundwater outside the excavation must circumnavigate the cut-off wall to reach the pumping wells inside the excavation [24,25].

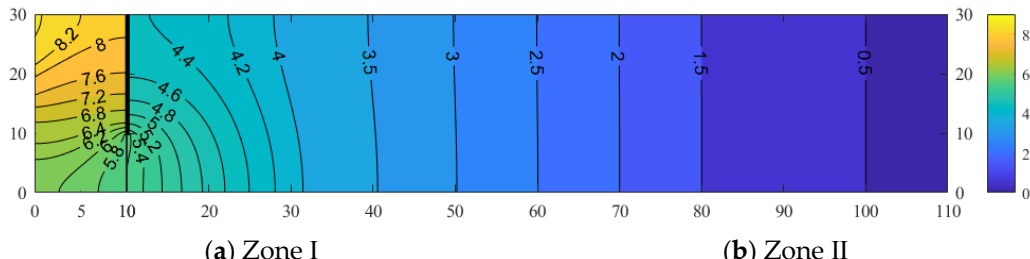

(**a**) Zone I          (**b**) Zone II

**Figure 4.** The contour of hydraulic drawdown for dewatering in the confined aquifer.

While the impermeable barrier restricts the inflow of groundwater from the external aquifer, a hydraulic connection between the interior and exterior is established via the opening beneath the wall. Although solution (7) for Zone I and solution (8) for Zone II are derived separately, the coupling between the two solutions has been successfully established by the discretization relationship (10) in our solving strategy. This can be observed from the result shown in Figure 4: the dominating horizontal flow in the external aquifer transforms into vertical flow along the cut-off wall and subsequently enters the excavation through the opening below the cut-off wall almost horizontally. This further demonstrates the correctness and validity of the approach proposed in this paper.

*5.2. PumpingTest Data*

To further validate the presented solution, pumping test data are employed for comparison. The excavation pit is located at the East Nanjing Road Station of Metro Line 10 in Shanghai. The excavation pit, which measures approximately 25 m in width and 152 m in length, can be approximated as a long-strip pit. The deposits at the site comprise various soil types, including fill, silty clay, muddy silty clay, muddy clay, clay, silty fine sand, silty clay, and silty fine sand. The first confined aquifer, AqI, mainly consists of silty fine sand

with a thickness of 11.0 m [22]. Lowering the water table in AqI is imperative to ensure safety during the excavation process. Diaphragm walling was established to function as a groundwater-proof curtain (cut-off wall), with the barrier inserted into the aquifer layer at depths ranging from 1.8 m to 10.8 m, averaging at 6.3 m in the calculations.

There are three dewatering wells (Y1 to Y3) used for groundwater lowering within the pit to depressurize AqI. The pumping rates for these wells are 130 m$^3$/d, 111.6 m$^3$/d, and 122.6 m$^3$/d, respectively. In the calculations, the total pumping rate of the three wells was averaged along the length of the excavation pit, resulting in an average of 2.5 m$^2$/d. Other calculation parameters, such as the horizontal hydraulic conductivity and vertical hydraulic conductivity, were obtained from [22]. The specific values of calculation parameters are listed in Table 2. Based on the solutions provided in this paper, calculations were performed for this conceptual model. Figure 5 presents hydraulic drawdown curves on both the inside and outside of the excavation pit along the top of the confined aquifer AqI. Figure 5 reveals a significant head difference between the inside and outside of the excavation pit due to the presence of the barrier wall. Furthermore, the calculation results were compared with the field data from two observation wells G1 and G2 (one located inside the excavation pit and one located outside the pit) [22] and they showed a good agreement. This further validates the reasonableness of the solutions presented in this paper.

**Table 2.** The calculation parameters in the pumping test example.

| Calculation Parameters | Value |
| --- | --- |
| Thickness of aquifer | 11 m |
| Depth of barrier insertion | 6.3 m |
| Spacing between cut-off walls | 25 m |
| Pumping rate | 2.5 m$^2$/d |
| Length of well screen | 7 m |
| Horizontal hydraulic conductivity | 5 m/d |
| Vertical hydraulic conductivity | 0.5 m/d |

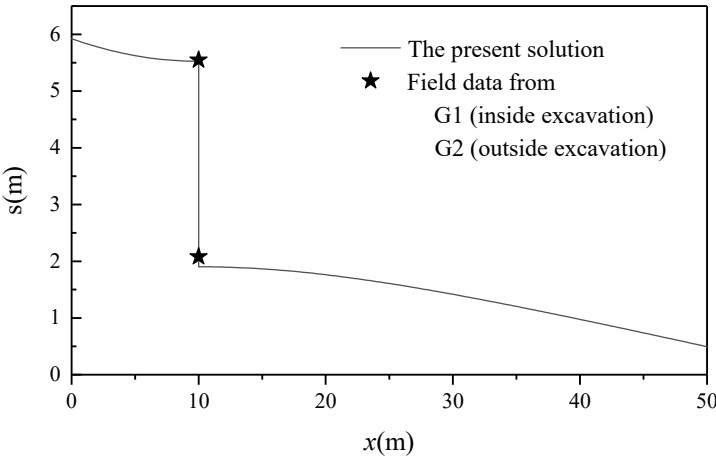

**Figure 5.** Hydraulic drawdown in the confined aquifer roof on the inside and outside of the excavation pit.

## 6. Results and Discussion

### 6.1. The Depth of the Cut-Off Wall into the Confined Aquifer

Figure 6 demonstrates the impact of the cut-off wall depth, $L_c$, on the dewatering process of the excavation. The parameter $L_c$ ranges from 0 m, representing the absence of a cut-off wall, to 30 m, which signifies the case of fully cutting off groundwater flow into the pit. From the figure, it can be observed that the dewatering trends at locations A and B, which are inside the excavation, significantly differ from that at location C, which is outside the excavation. The hydraulic drawdown inside the excavation increases with the length

of the cut-off wall into the confined aquifer. However, outside the excavation, the trend is reversed. This contrasting behavior is attributed to the barrier effects of the cut-off wall.

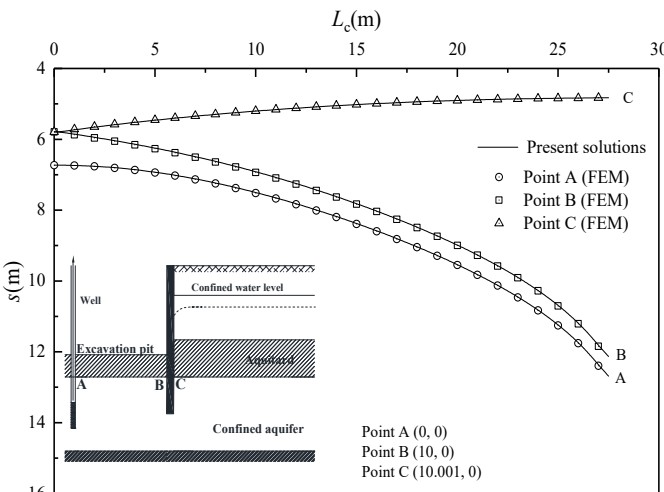

**Figure 6.** Drawdown results with varying depth of the cut-off wall.

For the confined water beneath the cut-off wall, although its seepage path is not directly blocked by the cut-off wall, there still exists a water level difference between the two sides of the pit. This is because in order for groundwater to flow into the pit around the lower portion of the cut-off wall, a head difference haves to occur between the inside and outside of the pit. As for the confined water above the cut-off wall, its flow direction is altered due to the obstruction of the cut-off wall. On one hand, it experiences a head difference as a result of this direct obstruction created by the cut-off wall, and on the other hand, an additional head difference is also needed for the outside water to flow into the excavation. In essence, the presence of the cut-off wall results in an increased water-level difference between the inside and outside of the excavation.

For comparison, Figure 7 illustrates the relationship between the pumping rate and the depth of cut-off wall into the confined aquifer, when the drawdown at the pumping well remains constant. As depicted in the figure, with an increase in the depth of cut-off wall, the pumping rates at locations A and B within the excavation gradually decrease. This trend highlights that a greater depth of the cut-off wall leads to a greater reduction in the pumping rate requirement. When a cut-off wall is in place, attaining a specific drawdown within the excavation becomes more feasible without causing extensive groundwater level reduction outside the pit in the confined aquifer. In other words, increasing the insertion depth of the cut-off wall into the aquifer will enhance the dewatering effectiveness of the pumping well while reducing the adverse effects of dewatering outside the excavation pit. The reduced hydraulic drawdown is beneficial for maintaining overall sustainability, as it helps mitigate potential environmental impacts and preserves the equilibrium of the confined aquifer system.

### 6.2. The Location and Length of the Partially Penetrating Well

Figure 8 presents the distribution of drawdown along the depth of the aquifer at the $x = 0$ cross-section, considering various well screen lengths $l_w$, ranging from 3 m to 15 m, under a constant pumping rate. The pumping well's starting position remains consistently at the uppermost part of the aquifer for this analysis. It is evident that the impact of the well screen is predominantly concentrated within the depth range of the impervious cut-off wall. Furthermore, an increase in well screen length leads to a more uniform drawdown distribution. Conversely, shorter well screens result in drawdown being concentrated near the bottom of the excavation pit, thereby achieving a more effective depressurization in the aquifer area beneath the exaction pit.

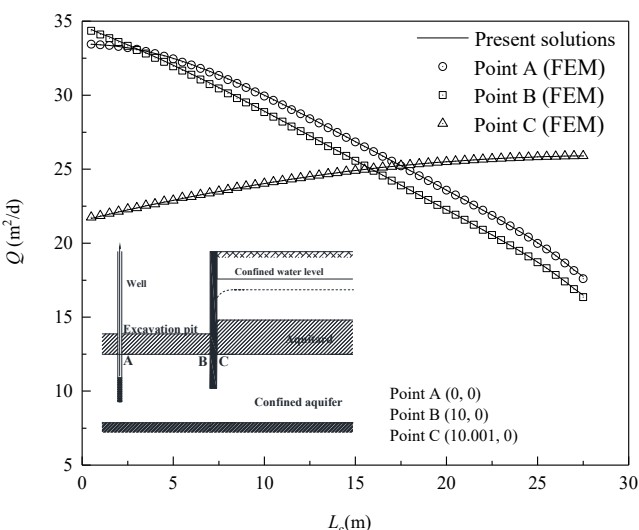

**Figure 7.** Relationships between the pumping rate and the depth of the cut-off wall under the same drawdown.

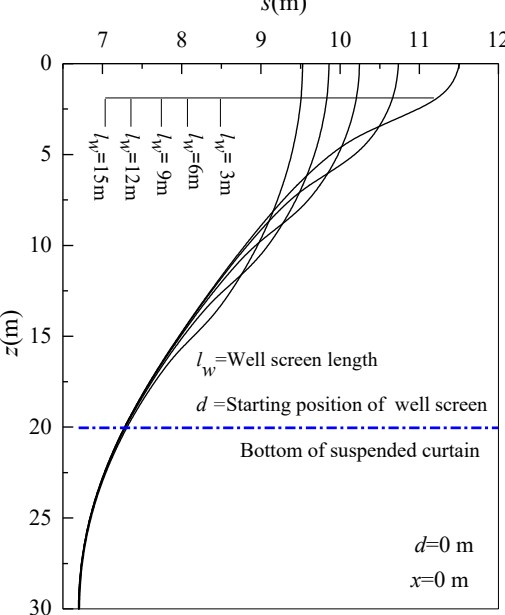

**Figure 8.** The effect of the length of the well screen on dewatering with the same pumping rate.

Figure 9 illustrates the impact of different pumping well depths ($d$ = 0 m, 2 m, 5 m, 12 m, 15 m) on the drawdown of the hydraulic level along the aquifer depth in Zone I ($x$ = 0 m). From the obtained results, it can be observed that the hydraulic drawdown decreases as the starting position $d$ increases. This indicates that when the pumping well is located closer to the top of the confined aquifer, more water can be extracted from Zone I. In contrast, in scenarios where the pumping well is situated directly above the base of the aquifer, (i.e., the case of $d$ = 15 m), its proximity to the gap under the cut-off wall results in the primary extraction of groundwater from the aquifer outside the excavation area. Consequently, this configuration produces the lowest drawdown within the excavation and imposes the most significant disturbance on outside aquifer. Overall, the closer and more concentrated the pumping well to the roof of the confined aquifer, the more effective the dewatering. This observation underscores the importance of placing pumping wells near the upper boundary of the confined aquifer in excavation dewatering practices, as it facilitates a more effective reduction of hydraulic head within the pit and minimizes water resource depletion due to the construction activities.

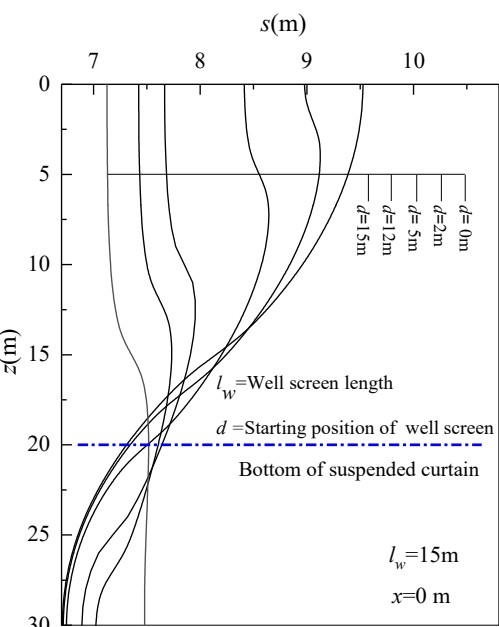

**Figure 9.** The effect of the location of the well on dewatering with the same pumping rate.

### 6.3. The Influence of Lateral Boundary Distance on Dewatering

The lateral head boundary constitutes one of the crucial hydrogeological factors that must be taken into account when dewatering in foundation excavations. Based on the provided solution, the drawdown distributions directly beneath the excavation pit ($z = 0$ m) within the pit, considering varying lateral boundary distances ($L$)—namely, $L = 20$ m, 40 m, 60 m, 80 m, and 100 m—are illustrated in Figure 10. It can be seen that the boundary distance has a significant impact on the dewatering effectiveness. Under the same pumping rate, as the lateral groundwater boundary approaches the excavation pit, leading to improved aquifer recharge, the dewatering effectiveness within the pit diminishes. Conversely, when the lateral groundwater boundary is farther away, the dewatering effectiveness within the pit improves. So, given that we are using fixed head boundaries as the lateral recharge conditions, a shorter distance allows for easier recharge. Consequently, the dewatering effectiveness of the excavation is also lower under such conditions. This also illustrates that if the excavation site is near water bodies such as reservoirs or lakes, the dewatering efficiency through the use of pumping wells alone will be limited. To enhance the effectiveness, it becomes necessary to employ a complete cut-off barrier to block hydraulic connections between the inside and outside of the pit.

### 6.4. The Impact of Discretizing the Common Boundary on Calculation Accuracy

The above numerical experiments indicate that the method developed in this study exhibits a high level of numerical accuracy, compared with FEM results. The good numerical performance can be attributed to the utilization of analytical techniques, i.e., the Finite Fourier transform method, in deriving the solutions. However, to address the heterogeneity caused by the cut-off barrier, a local discretization is introduced at the common boundary between the inside and outside of the pit in the solution. Hence, it is worthwhile to investigate the influence of this one-dimensional discretization on the numerical accuracy. In Figure 11, the results of the drawdown distribution in Zone Iare depicted, showcasing the impact of different discretization levels ($N = 1, 5, 10, 20$) along the segment cd. The results suggest a fast convergence regarding the amount of discretization. Additionally, it is worth noting that even when the partition employed is very rough ($N = 1$), the obtained results is still acceptable in terms of accuracy. This indicates the robustness and reliability of the method, as it still yields satisfactory results despite using a minimal level of subdivision.

The tolerance for rough discretizationis a result of the super convergence enjoyed by the Fourier method which is globally employed in the solutions developed in the study.

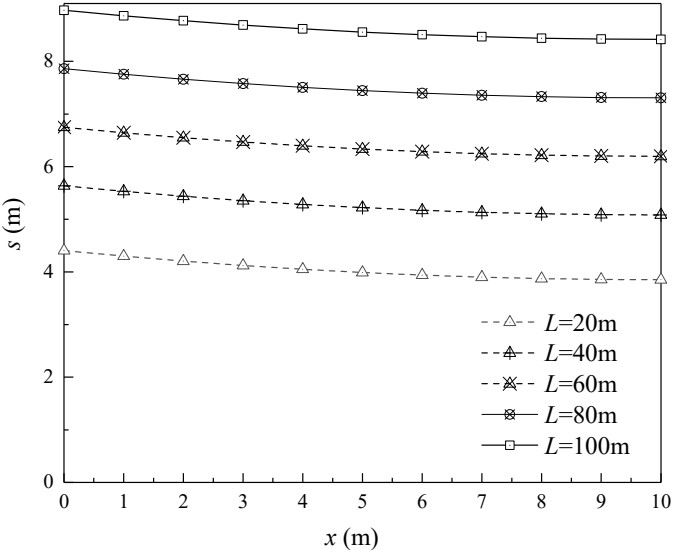

**Figure 10.** The drawdown curves along $z = 0$ min Zone I (inside the pit) with different lateral boundary distances.

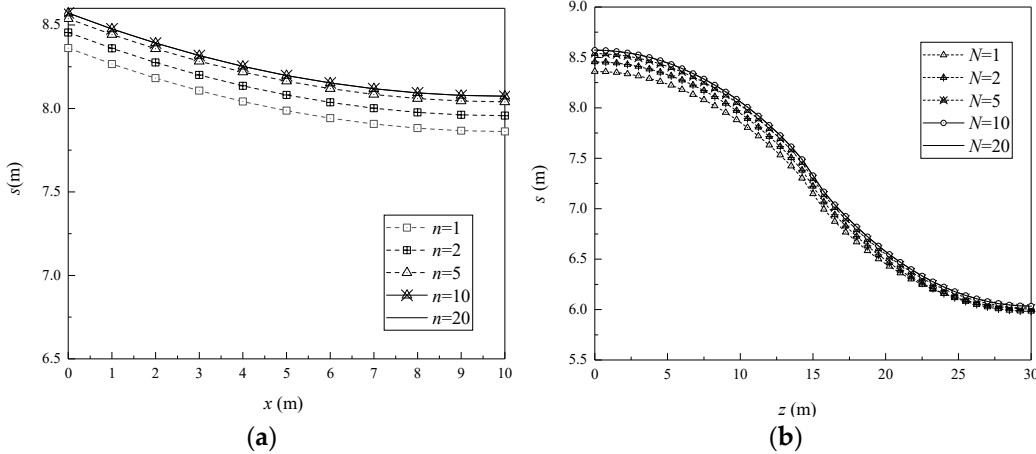

|     |     |
| :-: | :-: |
| (**a**) | (**b**) |

**Figure 11.** Drawdown distribution in Zone I under different levels of discretization: (**a**) along the cross-section $z = 0$ m and (**b**) along the cross-section $x = 0$ m.

## 7. Conclusions

This study focused on investigating the steady flow resulting from dewatering by a partially penetrating well in a confined aquifer with a cut-off barrier. Our analysis extended to consider the flow in both horizontal and vertical directions, incorporating both the barrier and the pumping well as flow boundary conditions. By establishing separate mathematical models for the inside and outside of the cut-off barrier and ensuring their interaction through continuous conditions, we were able to derive a semi-analytical solution for the problem. Based on the presented solution, the impact of the cut-off wall's embedment depth, the position and length of pumping wells, and the proximity to the lateral boundary was investigated, and the following conclusions were drawn:

(1)  To verify the effectiveness of our method, we compare our solution to the numerical results based on finite element method and field data from pumping test. The results of this study exhibit a good degree of consistency with the numerical simulations and field data. Our approach effectively characterizes the hydraulic difference between

the inside and outside of the pit, offering a comprehensive description of the barrier effect resulting from the cut-off wall.

(2)　The cut-off wall can effectively enhance the dewatering effect within the excavation, reduce the designed pumping volume, and minimize the adverse impact of dewatering on the external aquifer. This effect increases with the increasing embedment depth of the cut-off wall.

(3)　The closer and more concentrated the pumping wells are to the roof of the confined aquifer, the more effective the dewatering becomes. These findings indicate the significance of strategic well placement and cut-off wall design in excavation projects, not only for optimized dewatering but also for enhancing sustainability by minimizing groundwater resource depletion and reducing potential environmental impacts.

(4)　The results also imply that the proximity to the lateral constant-head boundary has a significant impact on dewatering effectiveness. Closer boundaries result in reduced effectiveness due to improved aquifer recharge.

(5)　The utilization of the Fourier method has demonstrated the remarkable convergence characteristics of the approach presented in this study. Even under a coarse discretization, the computed results consistently maintain a high level of approximation quality.

**Author Contributions:** Conceptualization, X.W. and W.T.; methodology, W.T. and X.W.; software, W.T. and H.K.; validation, H.K. and J.X.; formal analysis, W.T. and H.K.; investigation, H.K. and J.X.; resources, X.W.; data analysis, W.T. and H.K.; writing—original draft preparation, W.T.; writing—review and editing, J.X. and X.W.; visualization, W.T.; supervision, X.W. All authors have read and agreed to the published version of the manuscript.

**Funding:** This research was funded by the National Natural Science Foundation of China (grant number 41602284).

**Institutional Review Board Statement:** Not applicable.

**Informed Consent Statement:** Not applicable.

**Data Availability Statement:** The data that support the findings of this study are available from the corresponding author upon reasonable request.

**Conflicts of Interest:** The authors declare no conflict of interest.

## Appendix A. Solutions for Zone I and Zone II

To obtain the analytical solutions of the transformed problems (5) and (6), we process separately for two different cases of the transform parameter $n$: $n = 0$ and $n \geq 1$.

*Appendix A.1. Zone I*

(1)　when $n = 0$, Equations (5) become

$$\left\{ \begin{array}{l} \frac{\partial^2 \bar{s}_\mathrm{I}}{\partial x^2} = 0 \\ \frac{\partial \bar{s}_\mathrm{I}}{\partial x}\Big|_{x=0} = -\frac{Q_w}{K_x} \\ \frac{\partial \bar{s}_\mathrm{I}}{\partial x}\Big|_{x=B} = \int_{l_c}^{M} q(z)\mathrm{d}z \end{array} \right\} \tag{A1}$$

The equation in (A1) is a second-order ODE, with the general solution given by

$$\bar{s}_\mathrm{I} = c_1 x + c_2 \tag{A2}$$

By substituting the boundary conditions at $x = 0$ and $x = B$, we have

$$\begin{array}{l} c_1 = -\frac{Q_w}{K_x} \\ c_2 = C \end{array}$$

(2)    when $n \geq 0$, Equations (5) become

$$\left\{\begin{array}{l} \frac{\partial^2 \bar{s}_I}{\partial x^2} - \beta_n^2 \bar{s}_I = 0 \\ \frac{\partial \bar{s}_I}{\partial x}\Big|_{x=0} = -\frac{Q_w}{l_w K_x} \int_0^M \cos\left(\frac{\lambda_n z}{M}\right) dz = -\frac{Q_w}{K_x} \xi_n \\ \frac{\partial \bar{s}_I}{\partial x}\Big|_{x=B} = \int_{l_c}^M q(z) \cos\left(\frac{\lambda_n z}{M}\right) dz \end{array}\right\} \tag{A3}$$

which is also a second-order ODE, with the general solution given by

$$\bar{s}_I = c_1 \sinh(\beta_n x) + c_2 \cosh(\beta_n x) \tag{A4}$$

where

$$c_1 = -\frac{Q_w}{\beta_n K_x} \xi_n$$
$$c_2 = \frac{\cosh(\beta_n B)}{\sinh(\beta_n B)} \frac{Q_w}{\beta_n K_x} \xi_n + \frac{1}{\beta_n \sinh(\beta_n B)} \int_{l_c}^M q(z) \cos\left(\frac{\lambda_n z}{M}\right) dz$$

*Appendix A.2. Zone II*

We can derive the solutions for the Equation (6) in Zone II in a similar fashion.

(1)    when $n = 0$, Equation (6) becomes

$$\left\{\begin{array}{l} \frac{\partial^2 \bar{s}_{II}}{\partial x^2} = 0 \\ \frac{\partial \bar{s}_{II}}{\partial x}\Big|_{x=B} = \int_{l_c}^M q(z) dz \\ \bar{s}_{II}\big|_{x=L} = 0 \end{array}\right\} \tag{A5}$$

with the general solution given by

$$\bar{s}_{II} = c_1 x + c_2 \tag{A6}$$

where

$$c_1 = \int_{l_c}^M q(z) dz$$
$$c_2 = -L \int_{l_c}^M q(z) dz$$

(2)    when $n \geq 0$, Equation (6) becomes

$$\left\{\begin{array}{l} \frac{\partial^2 \bar{s}_{II}}{\partial x^2} - \beta_n^2 \bar{s}_{II} = 0 \\ \frac{\partial \bar{s}_{II}}{\partial x}\Big|_{x=B} = \int_{l_c}^M q(z) \cos\left(\frac{\lambda_n z}{M}\right) dz \\ \bar{s}_{II}\big|_{x=L} = 0 \end{array}\right\} \tag{A7}$$

which is a second-order ODE with the solution as follows

$$\bar{s}_{II} = c_1 \sinh(\beta_n x) + c_2 \cosh(\beta_n x) \tag{A8}$$

where

$$c_1 = \frac{\cosh(\beta_n L)}{\beta_n \cosh(\beta_n(L-B))} \int_{l_c}^M q(z) \cos\left(\frac{\lambda_n z}{M}\right) dz$$
$$c_2 = -\frac{\sinh(\beta_n L)}{\beta_n \cosh(\beta_n(L-B))} \int_{l_c}^M q(z) \cos\left(\frac{\lambda_n z}{M}\right) dz$$

By summarizing the above Equations (A2), (A4), (A6), and (A8), we have derived the general solutions, which are

$$\bar{s}_I = \left\{\begin{array}{ll} -\frac{Q_w}{K_x} x + C & n = 0 \\ \frac{\cosh(\beta_n(B-x))}{\beta_n \sinh(\beta_n B)} \frac{Q_w \xi_n}{K_x} & \\ + \frac{\cosh(\beta_n x)}{\beta_n \sinh(\beta_n B)} \int_{l_c}^M q(z) \cos\left(\frac{\lambda_n z}{M}\right) dz & n \geq 1 \end{array}\right\} \tag{A9}$$

$$\bar{s}_{\text{II}} = \left\{ \begin{array}{ll} (x-L)\int_{l_c}^{M} q(z)\mathrm{d}z & n = 0 \\ -\frac{\sinh(\beta_n(L-x))}{\beta_n \cosh(\beta_n(L-B))} \int_{l_c}^{M} q(z) \cos\left(\frac{\lambda_n z}{M}\right) \mathrm{d}z & n \geq 1 \end{array} \right\} \tag{A10}$$

**Appendix B. Solving Procedure for Unknown Function *q(z)***

If we partition the common boundary c-d (in Figure 2) into equal-length subintervals with endpoints $j$ =1,2,…,$N$, as shown in Figure A1, the integrals in Equation (9) can be approximated using the Riemann sums given by

$$\left. \begin{array}{l} \int_{l_c}^{M} q(z)\mathrm{d}z = \sum_{j=1}^{N} q_j \Delta z_j = -\frac{Q_w}{K_x} \\ \int_{l_c}^{M} q(z) \cos\left(\frac{\lambda_n z}{M}\right)\mathrm{d}z = \sum_{j=1}^{N} q_j \frac{2M}{\lambda_n} \cos\frac{\lambda_n z_j}{M} \sin\frac{\lambda_n \Delta z_j}{2M} = \sum_{j=1}^{N} q_j g_{nj} \end{array} \right\} \tag{A11}$$

where $z_j$ and $q_j$ represent the vertical coordinate and the flux at the endpoint $j$ respectively, and $g_{nj} = \frac{2M}{\lambda_n} \cos\frac{\lambda_n z_j}{M} \sin\frac{\lambda_n \Delta z_j}{2M}$.

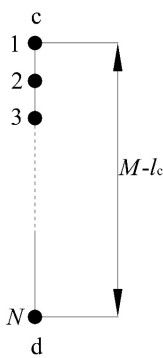

**Figure A1.** Local discretization of the common Boundary of c-d.

For each point $z_i$, the drawdown should satisfy condition (9), which results in a linear system with unknown qi as follows

$$[a_{ij}]\{q_j\} = \{b_i\} - \frac{C}{M} \tag{A12}$$

where

$$a_{ij} = \frac{2}{M} \sum_{n=1}^{\infty} \frac{1}{\beta_n} [\tanh(\beta_n(L-B)) + \coth(\beta_n B)] \cos\left(\frac{\lambda_n z_i}{M}\right) \cdot g_{nj} (i = 1, N; j = 1, N);$$

$$b_i = \frac{Q_w L}{M K_x} - \frac{2Q_w}{M K_x} \sum_{n=1}^{\infty} \frac{\xi_n \cos\left(\frac{\lambda_n z_i}{M}\right)}{\beta_n \sinh(\beta_n B)} (i = 1, N)$$

The Equation (A12) can be transformed into $\sum_{j=1}^{N} a_{ij} q_j + \frac{C}{M} = b_i$, which can be written in matrix form as

$$\begin{bmatrix} a_{11} & \cdots & a_{1N} & 1 \\ \vdots & \ddots & \vdots & \vdots \\ a_{N1} & \cdots & a_{NN} & 1 \end{bmatrix} \left\{ \begin{array}{c} q_1 \\ \vdots \\ q_N \\ \frac{C}{M} \end{array} \right\} = \left\{ \begin{array}{c} b_1 \\ \vdots \\ b_N \end{array} \right\} \tag{A13}$$

According to the Equation (A11), we also have

$$\int_{l_c}^{M} q(z)\mathrm{d}z = \sum_{j=1}^{N} q_j \Delta z_j = -\frac{Q_w}{K_x}$$

which in matrix form is

$$
\begin{bmatrix} \Delta z_1 & \cdots & \Delta z_N \end{bmatrix} \begin{Bmatrix} q_1 \\ \vdots \\ q_N \end{Bmatrix} = -Q_w/K_x \tag{A14}
$$

Finally, a combination of the Equations(A13) and (A14) yields the following linear system for the unknowns $q_i$ and $C$

$$
\begin{bmatrix} a_{11} & \cdots & a_{1N} & 1 \\ \vdots & \ddots & \vdots & \vdots \\ a_{N1} & \cdots & a_{NN} & 1 \\ \Delta z_1 & \cdots & \Delta z_N & 0 \end{bmatrix} \begin{Bmatrix} q_1 \\ \vdots \\ q_N \\ C/M \end{Bmatrix} = \begin{Bmatrix} b_1 \\ \vdots \\ b_N \\ -Q_w/K_x \end{Bmatrix} \tag{A15}
$$

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
