# Peer review of "Analysis of Steady Groundwater Flow in Confined Aquifer Due to Long-Strip Pit Dewatering with Suspended Cut-Off Wall"

_sustainability, doi:10.3390/su152215699_

Round 1
Reviewer 1 Report
Comments and Suggestions for Authors
1) To highlight the main formulas, the deduce process of the equations is suggest move to appendix.
2) Section 5 Verification: the numerical simulation model should be illustrated more clearly, including the model mesh, parameters, initial and boundary condition. Moreover finite model diagram is necessary.
3) The author compared the result obtained by the theoretical formula and numerical simulation to verify the computational accuracy of the theoretical formula. However the numerical simulation has the characteristic of uncertainty since it rely on the rational selection of parameters. The comparison between theoretical analysis and measured data in practical engineering is more convincing.
4) Figure 4: the result shown in Figure 4 obtained by numerical simulation or theoretical formula? Please introduce more clearly.
5) Figure 5 and 6: the schematic diagram of pit dewatering in Figure 5 and 6 is unclear, which should be improved
6) Figures: Application of different legends in the linear graphs can make these figures more understandable.
Reviewer 2 Report
Comments and Suggestions for Authors
- Assuming the problem is radial, it would be more appropriate to use r instead of the variable x. This is a common practice in many publications.
- for a better understanding, it would be appropriate to mark points a, b, c, d, e, as well as the coordinate system, on Figure 1, which are indicated in Figure 2.
- What scientific role does the water table indicated in Figures 1, 5, and 6 play in this study?
- Always use the same name for a given variable. For example, is it correct to say that 'flux function' is equal to 'flux distribution q(z)', or 'water level drawdown' is the same as 'drawdown' or 'hydraulic drawdown'. Similarly, this applies to 'cutoff curtain', which can be referred to as 'cut-off wall', 'waterproof curtain', 'cut-off barrier', 'impervious curtain', 'suspended curtain', 'water-proof wall' or simply 'curtain'.
- I would prefer to use the term 'Cut-off Wall' in the article because 'Cut-off Wall' denotes a stronger and more permanent underground sealing barrier, typically made from durable materials such as concrete or steel. In contrast, 'Curtain' often implies a temporary or less robust barrier, often made from geotextiles or similar materials, and might be suitable for occasional or short-term use. In your article, opting for 'Cut-off Wall' would be more appropriate if you want to emphasize the durability and stability of the sealing structure.
- When introducing the coordinate system, it is essential to specify that the well is at the origin of the coordinate system as it is located at the center of the excavation pit.
- At the stage of examining Figure 2 on the left, the reader cannot discern the significance of the points numbered 1, 2, 3, ..., N, where point c corresponds to point 1 and point d corresponds to point N. Additionally, it is unclear why the distance (=M-lc??) between points c and d, marked as 1 and N, has been defined, and what its value represents
- Please explain to the reader how the confined water level, as depicted in Figure 1, is related to the unknown hydraulic drawdown sII in Zone II
- Could you indicate course of the sI and sII on the Figure 1 or 2?
- It would have been appropriate to include a reference to the relevant literature where readers could learn how the boundary value problem (1) and (2) was derived.
- It would be appropriate to specify the dimension of each mentioned parameter in the system of equations (1) and (2). Usually, the pumping rate is expressed in cubic meters per day (m³/d) or cubic meters per second (m³/s). Please explain the conversion to square meters per day (m²/d)
- The meaning of the term 'flux distribution' is not clearly defined.
Reviewer 3 Report
Comments and Suggestions for Authors
This paper is considered as a quite interesting scientific work regarding the study of the steady flow resulting from dewatering by a partially penetrating well in a confined aquifer with a cut-off barrier. The subject is within the topics of the Sustainability Journal. The manuscript is clearly written following a structure that contains analysis results documented and presented in an informative, reliable and explanatory way. My recommendation is that the manuscript should be accepted for publication in its present form.
Reviewer 4 Report
Comments and Suggestions for Authors
General Comments:
- How does the choice of well depth, the placement of the cut-off wall, and proximity to lateral head boundaries impact the required pumping rate and drawdown of the confined water level within an excavation?
- How accurately and effectively the continuous conditions and mathematical methods like the finite Fourier cosine transform and boundary transformation can capture the complex interplay between the two zones in a confined aquifer?
- To enhance the practical applicability of the study’s findings, conducting field studies to validate the conclusions in real excavation projects is recommended. This would provide concrete evidence of the effectiveness of strategic well placement and cut-off wall design under varying geological and hydrogeological conditions.
Sec 2.1. Zone I
What are the specific characteristics of the lateral boundaries of Zone I, and how do these conditions influence the formulation of the boundary value problem for describing steady-state groundwater flow within Zone I in both horizontal and vertical directions?
Sec 3. Fourier transform method
How does the introduction of geometric regularity by partitioning the barrier and treating it as a no-flow boundary affect the analytical tractability of addressing groundwater flow in comparison to the original scenario with heterogeneously embedded barriers in the aquifer?
5. Verification
Explain the specific areas where the mesh was refined. Justify why these regions were chosen and how it contributes to computational accuracy.
Figure 4.
What specific insights can be gathered from Figure 4 in terms of hydraulic behavior, the influence of the waterproof barrier, and the interaction between Zones I and II in the context of groundwater flow?
Conclusions:
As a future recommendation, it is advisable to work on the development of standardized sustainability metrics and guidelines tailored to excavation dewatering. This will enable a more consistent evaluation and comparison of environmental impacts and efforts to conserve resources.
Round 2
Reviewer 4 Report
Comments and Suggestions for Authors
Nil.